# High PANX1 Expression Leads to Neutrophil Recruitment and the Formation of a High Adenosine Immunosuppressive Tumor Microenvironment in Basal-like Breast Cancer

**DOI:** 10.3390/cancers14143369

**Published:** 2022-07-11

**Authors:** Wuzhen Chen, Baizhou Li, Fang Jia, Jiaxin Li, Huanhuan Huang, Chao Ni, Wenjie Xia

**Affiliations:** 1Department of Breast Surgery (Surgical Oncology), Second Affiliated Hospital, Zhejiang University School of Medicine, Hangzhou 310009, China; chenwuzhen@zju.edu.cn (W.C.); jiaf@zju.edu.cn (F.J.); 22018203@zju.edu.cn (J.L.); huanghuanhuanth@163.com (H.H.); 2Key Laboratory of Tumor Microenvironment and Immune Therapy of Zhejiang Province, Hangzhou 310009, China; 3Cancer Institute, Zhejiang University, Hangzhou 310009, China; 4Department of Pathology, Second Affiliated Hospital, Zhejiang University School of Medicine, Hangzhou 310009, China; alexlibz@gmail.com; 5General Surgery, Cancer Center, Department of Breast Surgery, Zhejiang Provincial People’s Hospital (Affiliated People’s Hospital, Hangzhou Medical College), Hangzhou 310014, China

**Keywords:** pannexin 1 (PANX1), neutrophils, adenosine, tumor microenvironment (TME), breast cancer

## Abstract

**Simple Summary:**

A high adenosine level is an important characteristic of the tumor microenvironment (TME) in breast cancer. Pannexin 1 (PANX1) can release intracellular ATP to the extracellular space and elevate extracellular ATP (exATP) levels under physiological conditions. PANX1 has been found to be a poor prognostic factor in breast cancer, however, the role of PANX1 in breast cancer remains unknown. In this study, we performed RNA sequencing, bioinformatics analysis, surgical specimen histological validation, and exATP/extracellular adenosine (exADO) assays to reveal the role of PANX1 in regulating the immune microenvironment of basal-like breast cancer. The results revealed that PANX1 acted as a poor prognostic factor for breast cancer and had high expression in basal-like breast cancer. PANX1 expression was positively correlated with exATP and exADO levels in basal-like breast cancer TME. PANX1 expression was also positively correlated with tumor-associated neutrophil (TAN) infiltration in breast cancer TME, and TANs highly expressed CD39/CD73, which synergistically build a high exADO immunosuppressive TME and promote tumor progression. This study suggests that high PANX1 expression is associated with high TAN infiltration and adenosine production to induce local immunosuppression in basal-like breast cancer TME.

**Abstract:**

**Background:** A high adenosine level is an important characteristic of the tumor microenvironment (TME) in breast cancer. Pannexin 1 (PANX1) can release intracellular ATP to the extracellular space and elevate extracellular ATP (exATP) levels under physiological conditions. **Methods**: We performed public database bioinformatics analysis, surgical specimen histological validation, RNA sequencing, and exATP/extracellular adenosine (exADO) assays to reveal the role of PANX1 in regulating the immune microenvironment of basal-like breast cancer. **Results**: Our results revealed that PANX1 acted as a poor prognostic factor for breast cancer and had high expression in basal-like breast cancer. PANX1 expression was positively correlated with exATP and exADO levels in basal-like breast cancer TME. PANX1 expression was also positively correlated with tumor-associated neutrophil (TAN) infiltration in breast cancer TME and TANs highly expressed ENTPD1 (CD39)/NT5E (CD73). **Conclusions**: This study suggests that high PANX1 expression is associated with high TAN infiltration and adenosine production to induce local immunosuppression in basal-like breast cancer TME.

## 1. Introduction

Breast cancer is the most common malignant tumor in women. Basal-like breast cancer, as an important breast subtype, is a heterogeneous group of tumors defined by negative immunohistochemical staining for estrogen receptor (ER) and progesterone receptor (PR) and a lack of overexpression of human epidermal growth factor receptor 2 (HER2) with different levels of expression in basal cell keratins and myoepithelial markers [1,2]. Basal-like breast cancer is prone to recurrence and metastasis and has a poor prognosis due to the lack of specific treatments [3,4]. According to the PAM50 algorithm, 71% of triple negative breast cancer (TNBC) was found to be basal-like breast cancer [2]. Basal-like breast cancer is a clinically exclusive diagnosis that needs to be more precisely characterized at the molecular level. Compared with other subtypes, the immune status of the tumor microenvironment (TME) has a significant impact on the treatment and prognosis in basal-like breast cancer [5,6]. Therefore, it is of clinical importance to investigate the key regulatory genes related to the immune TME of basal-like breast cancer.

Pannexin 1 (PANX1), a member of the gap junction protein family, mediates the release of intracellular ATP to the extracellular microenvironment in its full-length form [7]. Extracellular ATP (exATP) and its metabolite extracellular adenosine (exADO) are important factors that regulate local immune TME [8]. Under physiological conditions, exATP released by PANX1 promotes innate and adaptive immune responses by attracting immune cells [9]. In the TME, this process is disrupted, and exATP is rapidly metabolized by nucleotidases ENTPD1 (CD39) and NT5E (CD73) to generate exADO [10]. In the TME, exADO is a key factor that contributes to local immunosuppression [11]. Recently, PANX1 expression was found to be important in suppressing airway inflammation in the asthma mouse model, and the knockdown of PANX1 resulted in increased airway inflammation [12]. In breast cancer, PANX1 was overexpressed and promoted the transformation of tumor cells to the epithelial-to-mesenchymal transition (EMT) phenotype; breast cancer patients with high PANX1 expression had a poor clinical prognosis [13]. However, the way in which PANX1 affects tumor-infiltrating immune cells (TIICs) and the immune TME by regulating exATP levels has not been reported.

This study revealed that basal-like breast cancer tissues had high PANX1 expression, which was positively correlated with tumor-associated neutrophils (TANs) and the accumulation of exADO, forming an immunosuppressive TME.

## 2. Materials and Methods

### 2.1. Data Acquisition

The TCGA-BRCA RNA-Seq gene expression matrix data with clinical information were downloaded from the TCGA data portal (https://portal.gdc.cancer.gov/) (accessed on 5 April 2021) and normalized using R package TCGAbiolinks [14,15,16]. METABRIC [17] microarray (Illumina HT-12 v3) normalized gene expression data were downloaded from the cBioPortal website (https://www.cbioportal.org) (accessed on 7 April 2021) [18]. The GSE103091 dataset (normalized gene expression data, GPL570, Affymetrix Human Genome U133 Plus 2.0 Array) [19,20] was downloaded from the GEO database (https://www.ncbi.nlm.nih.gov/geo/query/acc.cgi?acc=GSE103091) (accessed on 12 April 2021) using R package GEOquery for further CIBERSORT analysis. The METABRIC and TCGA-BRCA samples were classified into molecular subtypes using the PAM50 algorithm [21] using R package genefu [22].

### 2.2. Clinical Specimen Collection

Formalin-fixed paraffin-embedded (FFPE) sections from 12 TNBC patients who underwent surgery at the Second Affiliated Hospital of Zhejiang University School of Medicine (SAHZU) were collected from March 2020 to December 2020. Fresh surgical tumor specimens from 21 patients with TNBC and 12 patients with Luminal A cancer who underwent surgery at SAHZU were collected from March 2020 to April 2022 (with paired peripheral blood samples in 6 patients). Clinicopathological and survival information was also collected after receipt of informed consent and approval from the ethics committee. Clinical baseline characteristics of the included patients and corresponding experimental procedures for the surgical specimens have been summarized in Table 1, and detailed patient information was included in Appendix A. The percentage of stromal tumor-infiltrating lymphocytes (TILs) in breast cancer was evaluated under recommendations of the International TILs Working Group [23].

### 2.3. Cell Lines and Culture Conditions

The MDA-MB-231, HCC-1937, and MCF-7 human breast cancer cell lines were all obtained from the American Type Culture Collection (ATCC, Manassas, VA, USA). MDA-MB-231 and MCF-7 cells were cultured in DMEM with 10% fetal bovine serum, while HCC-1937 cells were cultured in RPMI 1640 medium with 10% fetal bovine serum. All cell lines were grown with 5% CO_2_ at 37 °C.

### 2.4. RNA Sequencing

Total RNA was isolated using Trizol (Invitrogen, Carlsbad, CA, USA) and RNeasy mini kit (Qiagen, Valencia, CA, USA) according to the manufacturer’s protocol. For clinical samples, single-end libraries were subsequently constructed using the standard protocol provided by BGI (BGI, Shenzhen, China) and were then sequenced on the BGISEQ-500 platform. Clinical sample RNA sequencing was performed in 19 fresh surgical samples (TNBC: 16; Luminal A: 3) and tumor-associated neutrophils with paired peripheral blood neutrophils sorted from 6 TNBC patients (Table 1). For cell lines, RNASeq library was prepared using Illumina TruSeq RNA sample preparation kit (Illumina, San Diego, CA, USA) and RNA sequencing was performed by Illumina HiSeq 2500 platform in MDA-MB-231 and HCC-1937 cells (WT/shPANX1/shCTRL).

### 2.5. Bulk Transcription Data Analysis

The relative proportions of tumor-infiltrating immune cells were inferred using TIMER (TCGA-BRCA data and own data) [24,25] (V1: https://cistrome.shinyapps.io/timer; V2: http://timer.comp-genomics.org/) (accessed on 21 March 2021) and CIBERSORT LM22 (TCGA-BRCA data) (accessed on 9 April 2021) [26] (22 immune cell reference profiles: https://cibersort.stanford.edu/). For CIBERSORT processed data relating to neutrophil infiltration, we screened for and removed all outliers (2 standard deviations above or below the mean) for further linear regression analysis. For RNA-seq data of fresh surgical specimens, the fractions of tumor-infiltrating immune cells were inferred using TIMER V2 and quanTIseq [27] (http://timer.comp-genomics.org/) (accessed on 3 May 2021). GEPIA2 [28] was used to detect the top 250 genes related to PANX1 (https://gepia2.cancer-pku.cn/) (accessed on 2 May 2021). A list of human immune-related genes was derived through the Immunology Database and Analysis Portal (ImmPort) [29] (https://www.immport.org/) (accessed on 2 May 2021). The Lehmann’s TNBC typing had been adopted to evaluate the PANX1 expression in different TNBC subtypes by using the webtool TNBCtype [30,31] (https://cbc.app.vumc.org/tnbc/) (accessed on 12 April 2022). The enriched gene ontology (GO) immune-related pathways were identified via the ClueGO v2.5.8 plugin in Cytoscape 3.8.2 software [32,33]. Using TCGA-BRCA and METABRIC data with clinical information, the overall survival (OS) status in different PAM50 subtypes of breast cancer under PANX1 high/low expression (median as the cut-off) was analyzed by R package Survminer and Survival. All packages used in this study were run in R environment 4.0.5.

### 2.6. Single Cell Transcription Data Analysis

The TNBC single-cell dataset [34] was downloaded from the Broad Institute Single Cell Portal (https://singlecell.broadinstitute.org/single_cell/study/SCP1106/) (accessed on 7 September 2021). The dataset was analyzed using the Seurat R package. Nonlinear dimensional reduction (tSNE) was used to visualize clustering results. Epithelial cell clusters with highly variable CNVs were determined to be malignant by the inferCNV R package. The SingleR R package was used for cell type annotation, and the default annotated file (Wu_EMBO_metadata.csv) was used as a reference. The single-cell dataset was divided into PANX1 high and low expression groups according to the PANX1 expression of each tumor sample in the count matrix. We calculate the average PANX1 expression for each tumor sample and choose the median value as the cut-off (PANX1 low expression tumor group: CID4513, CID4515, and CID44971 (included cell number = 9623); PANX1 high expression tumor group: CID44041 and CID44991 (included cell number = 4252)).

### 2.7. Immunofluorescence Staining

Gene colocalization was validated by monoclonal antibody-based immunofluorescence. FFPE sections were subjected to antigen retrieval by heating the slides in citrate buffer for 2 min, after which they were incubated with primary antibodies (anti-human PANX1 (#AB139715, 1:500, Abcam, Cambridge, MA, USA) and anti-human MPO (#AF3667, 1:300, R&D Systems, Minneapolis, MN, USA) antibodies) at 4°C overnight. Then, the slides were incubated with fluorescein-labeled secondary antibodies (Cy3-conjugated donkey anti-rabbit IgG (#AP182C, 1:200, Merck, Darmstadt, Germany), Alexa Fluor 488-conjugated donkey anti-goat IgG (#A11055, 1:100, ThermoFisher, Waltham, MA, USA)) at room temperature, stained with DAPI, and photographed under a laser confocal microscope (OLYMPUS IX83-FV3000-OSR).

### 2.8. Immunohistochemical Staining

Gene expression was validated by monoclonal antibody-based immunohistochemistry. Immunohistochemical staining of FFPE slides, which were deparaffinized and rehydrated before the antigen retrieval step, was performed using the Envision method. Endogenous peroxidase was blocked by incubating the slides with 3% H_2_O_2_. FFPE slides were heated in citrate buffer for 2 min, incubated with primary antibodies (rabbit anti-human PANX1(#AB139715, 1:500, Abcam, Cambridge, MA, USA)/mouse anti-human ENTPD1 (clone A1, 1:300, Biolegend, San Diego, CA, USA)/anti-human NT5E antibodies (clone AD2, 1:300, Biolegend, San Diego, CA, USA)) at 4 °C overnight, and then incubated with the secondary antibody (HRP-conjugated goat anti-mouse IgG, #AB205719, 1:2000, Abcam; HRP-conjugated goat anti-rabbit IgG, AB205718, 1:5000, Abcam, Cambridge, MA, USA) for 30 min at room temperature. 3,3′-Diaminobenzidine (DAB) chromogen (Zhongshan Jinqiao Biotech, Beijing, China) was used for visualization. The intensity and frequency were used as evaluation indexes based on PANX1 staining. The expression intensity was divided into 4 subgroups: negative (0), weakly positive (1), positive (2), and strong positive (3). The expression frequency was divided into 5 subgroups: 0–10% (1), 11–30% (2), 31–50% (3), 51–75% (4), and 76–100% (5). PANX1 comprehensive score = intensity*frequency. The TNBC specimens included in immunofluorescence and immunohistochemical staining were subjected to transcriptomic analysis and confirmed to be basal-like subtype using the PAM50 algorithm (Table 1). The samples with the upper 50% of PANX1 comprehensive scores were defined as the PANX-high group (n = 3), while the samples with the lower 50% of PANX1 comprehensive scores were defined as the PANX-low group (n = 3). We quantitatively assessed the ENTPD1 and NT5E protein expression levels by ImageJ FIJI software [35] using the method developed by Alexandra Crowe and Wei Yue [36].

### 2.9. Neutrophil Isolation

Fresh surgical specimens were cut into small pieces and digested in medium containing 1 mg/mL collagenase IV (#V900893, Sigma-Aldrich, Saint Louis, MO, USA) at 37 °C in a constant temperature shaker for 2 h. The cell suspension was filtered through 40 μm nylon mesh (BD FALCON, #352340) for subsequent detection or culture. The separation of neutrophils from surgical specimens and peripheral blood was performed with a human neutrophil isolation kit according to the manufacturer’s protocol (Miltenyi Biotec, Auburn, CA, USA). For the neutrophils RNA-seq data, key immune-related gene expression was analyzed, and the heatmap was generated by the ComplexHeatmap R package.

### 2.10. ShRNA Knockdown of PANX1

MDA-MB-231 and HCC-1937 cells were transfected using the PANX1 human shRNA plasmid kit (OriGene, Rockville, MD, USA) (sequence: 5′-CGCAATGCTACTCCTGACAAACCTTGGCATGTCAAGAGCATGCCAAGGTTTGTCAGGAGTAGCATTGTT-3′). The PANX1/ENTPD1/NT5E expression was evaluated by RNA sequencing using Illumina HiSeq 2500 platform in MDA-MB-231 and HCC-1937 cells (WT/shPANX1/shCTRL). Single-cell colonies of PANX shRNA-expressing cells were selected with puromycin and examined for PANX1 knockdown. Stable knockdown samples showed a 70–90% reduction in PANX1 expression. Cells were maintained under puromycin selection pressure and were periodically examined for effective PANX1 knockdown using western blot.

### 2.11. Extracellular ATP/ADO Assay

In 24-well plates, 2 × 10^4^ cells (MDA-MB-231/MDA-MB-231 shPANX1/MDA-MB-231 shControl/HCC-1937/HCC-1937 shPANX1/HCC-1937 shControl/MCF-7) per well were allowed to adhere overnight after which they were removed by centrifugation at 4 °C for 10 min (MCF-7 as a representative of luminal breast cancer cell lines). For probenecid (PRB) treatment group, the MDA-MB-231/HCC-1937 cells were pretreated for 10 min with 1 mM PRB (#P8761, Sigma-Aldrich, Saint Louis, MO, USA). The supernatant was then collected for ATP/ADO detection assays. Fresh surgical TNBC tissues and normal breast tissue specimens were cut into small pieces and digested in medium containing 1 mg/mL collagenase IV (#V900893, Sigma-Aldrich, Saint Louis, MO, USA) at 37 °C at a constant temperature. The cell suspension was then filtered through 40 μm nylon mesh (#352340, BD FALCON) for subsequent ATP/ADO detection assays. ATP/ADO levels were measured using the ATP/ADO Assay Kit (fluorometric) (Abcam, Cambridge, MA, USA) according to the manufacturer’s protocol.

### 2.12. Statistical Analyses

Statistical significance was determined using unpaired two-tailed Student’s *t*-tests or analysis of variance (ANOVA) followed by Tukey’s test. Pearson’s correlation and Spearman’s rank correlation statistics were used to determine the correlation for linear regression. A log-rank test was performed to assess PANX1 as a survival biomarker. For all in vitro assays, data are representative of at least three independent experiments, which each included three technical replicates unless otherwise stated. Differences in PANX1 expression in different subtypes were assessed using one-way ANOVA (Tukey’s test). Paired differences in ATP/ADO levels and ENTPD1/NT5E expression between different groups were assessed using two-tailed *t*-tests. Statistical analyses were performed using GraphPad Prism software (version 9.0) and R software (version 4.0.5, R Core Team, http://www.R-project.org/) (accessed on 5 April 2021). The results are given as mean ±  S.D. and *p* values < 0.05 were considered significant (unless otherwise stated). The details of the statistical analysis are provided in the results section and figure legends.

## 3. Results

### 3.1. PANX1 Was Highly Expressed in Basal-like Breast Cancer

Compared with normal tissues, we found that PANX1 was highly expressed in breast cancer (BRCA), colon adenocarcinoma (COAD), esophageal carcinoma (ESCA), head and neck squamous cell carcinoma (HNSC), kidney chromophobe (KICH), lung adenocarcinoma (LUAD), lung squamous cell carcinoma (LUSC), and stomach adenocarcinoma (STAD) based on the web tool TIMER (TCGA-BRCA data) (*p* < 0.001 was significant) (Figure 1A). Using TCGA-BRCA (n = 1083) and METABRIC (n = 1699) data, high PANX1 expression suggested poor prognosis for basal-like breast cancer in terms of overall survival (OS) (*p* < 0.05), while not for Luminal A, Luminal B, HER2-enriched, or normal-like subtype (PAM50 algorithm) (Figure 1B and Appendix A). To further investigate the role of PANX1 expression in breast cancer, we analyzed TCGA-BRCA and METABRIC transcriptome data and confirmed that PANX1 was highly expressed in the basal-like subtype (TCGA-BRCA (n = 1083) and METABRIC (n = 1699) data, PAM50 algorithm) (Figure 1C,D). We also explored PANX1 expression in different TNBC subtypes (Lehmann’s TNBC typing; TCGA-BRCA-TNBC (n = 157) and METABRIC-TNBC (n = 158) data) and found there was no significant difference of PANX1 expression among different TNBC subtypes (*p* > 0.05) (Appendix A). Using TCGA-BRCA data, we further explored the relationship between PANX1 expression and tumor stage and found no significant differences in PANX1 expression across stages in all breast cancer samples or basal subtype (*p* > 0.05) (Appendix A). The above results suggested that, as a poor prognostic factor in breast cancer, PANX1 was highly expressed in basal-like breast cancer.

### 3.2. PANX1 Expression Positive Correlated with ENTPD1/NT5E Expression in the TME

PANX1 is acknowledged as a dominant regulator of exATP release, while exATP and its metabolite exADO are believed to induce an immune-suppressive TME [10]. An analysis of TCGA-BRCA and METABRIC basal-like subtype transcriptome data revealed that the group with high expression of PANX1 (top 50%) had higher ENTPD1 and NT5E expression levels than the group with low expression (bottom 50%) (*p* < 0.001, TCGA-BRCA (n = 186) and METABRIC (n = 199)) (Figure 2A). Further linear analysis suggested a positive correlation between PANX1 expression and ENTPD1 (R^2^ = 0.47 (TCGA) and 0.14 (METABRIC); *p* < 0.001) and NT5E (R^2^ = 0.39 (TCGA) and 0.08 (METABRIC); *p* < 0.001) expression in TCGA-BRCA and METABRIC basal-like subtype transcriptome data (Figure 2B). The RNA-seq data of surgical breast cancer specimens (basal-like PANX1 high: n = 6; basal-like PANX1 low: n = 6; Luminal subtype: n = 3) suggested a higher expression of ENTPD1 and NT5E in the basal-like PANX1-high group compared to basal-like PANX1-low and Luminal group (*p* < 0.05) (Figure 2C). According to immunohistochemistry of basal-like surgical specimens, the expressions of ENTPD1 and NT5E were higher in the PANX1 high group (n = 3) than in the PANX1 low group (n = 3) (Table 2, Figure 2D and Appendix A). The above results indicated that PANX1 expression was positive correlated with ENTPD1 and NT5E expression in the basal-like breast cancer TME.

### 3.3. PANX1 Expression Was Positively Correlated with TAN Infiltration in Basal-like Breast Cancer

Using CIBERSORT LM22, we analyzed the effect of PANX1 expression on basal-like breast cancer immune microenvironment in TCGA-BRCA (n = 186) data and GSE103091(n = 238) dataset. We found that the abundances of infiltrating neutrophils (*p* < 0.05), resting memory CD4^+^ T cells (*p* < 0.05), follicular helper T cells (*p* < 0.05), monocytes (*p* < 0.05), CD8^+^ T cells (*p* < 0.05), and activated natural killer (NK) cells (*p* < 0.05) were significantly different between PANX1 high expression (top 50%) and low expression (bottom 50%) tumors (Figure 3A). Coexpressed PANX1-related genes were obtained using GEPIA2; the immune-related GO analysis suggested that PANX1 and its coexpressed genes were related to granulocyte migration, neutrophil activation, etc. (*p* < 0.01) (Figure 3B). Furthermore, the relationship between PANX1 expression and TIICs was assessed using TIMER in the TCGA-BRCA basal-like subtype data, and a positive correlation was indicated between PANX1 and the infiltration level of TANs and CD4^+^ T cells (*p* < 0.05) (Figure 3C). Using TCGA-BRCA data and the CIBERSORT algorithm, we verified the positive correlation between PANX1 expression and TANs infiltration in the basal-like subtype (n = 25, outliers have been filtered; R^2^ = 0.32; *p* = 0.003; Pearson’s correlation) (Figure 3D).

In addition, the PANX1 expression was positively correlated with myeloperoxidase (MPO) expression in the TCGA-BRCA basal-like subtype (R^2^ = 0.19; *p* < 0.001) (Figure 3E). Moreover, the coexpression of PANX1 and MPO in basal-like breast cancer paraffin-embedded surgical specimens was also observed using immunofluorescence (Figure 3F). PANX1 and MPO expression was positive correlated, indicating that high PANX1 expression might promote TAN infiltration in basal-like breast cancer TME. However, the way in which PANX1 establishes an immunosuppressive microenvironment with TANs should be further explored.

### 3.4. Immunosuppressive TANs Demonstrated More Infiltration in Basal-like Breast Cancer with High PANX1 Expression

quanTIseq and TIMER deconvolution methods were reported to have high deconvolution performance for RNA-seq data from different tumor types and could be suitable tools for the further exploration of tumor-infiltrating neutrophils [37]. RNA-seq data from fresh surgical specimens were converted into infiltrating immune cell information by quanTIseq and TIMER. The results revealed that basal-like breast cancer with high PANX1 expression (n = 6) had more infiltrating TANs than basal-like breast cancer with low PANX1 expression (n = 6) and Luminal subtype (n = 3) (*p* < 0.05 for TIMER and quanTIseq method) (Figure 4A–C). In addition, the relationship between ENTPD1/NT5E expression and the infiltration level of TANs in basal-like subtype was evaluated. In TCGA-BRCA data, ENTPD1/NT5E expression was positively correlated with the infiltration level of TANs in basal-like subtype (*p* < 0.05) (Figure 4D: CIBERSORT method; Figure 4E: TIMER method). Neutrophils were purified from fresh basal-like breast cancer surgical specimens (n = 6) and paired peripheral blood samples (n = 6). Transcriptome analysis revealed TANs had higher expressions of nucleotidases ENTPD1 and NT5E (*p* < 0.05) and higher expressions of immunosuppressive cell recruitment-related cytokines CCL2 and CCL17 (*p* < 0.05) compared with peripheral blood neutrophils (Figure 4F). The above results suggested that PANX1 expression was positively associated with the TANs infiltration in TME, and TANs could convert exATP to exADO by highly expressing ENTPD1 and NT5E.

### 3.5. High PANX1 Expression Induced a High exADO Immunosuppressive TME in Basal-like Breast Cancer

To clarify whether high PANX1 expression in basal-like breast cancer could establish an immunosuppressive TME with local high exADO levels, we measured exATP and exADO levels in the supernatant of cultured breast cancer cell lines and found that exATP and exADO levels in MDA-MB-231 and HCC-1937 (basal-like subtype cell lines) cell culture media were significantly higher than those in MCF-7 (Luminal subtype cell line) cell culture media (n = 19 for each group; *p* < 0.05) (Figure 5A). Knocking down PANX1 and probenecid treatment in MDA-MB-231 and HCC-1937 cell lines led to a downregulation of exATP and exADO levels in the cell culture media (n = 19 for each group; *p* < 0.05) (Figure 5A). Moreover, exATP and exADO levels in digested tissue supernatant of basal-like breast cancer surgical samples were significantly higher than those in Luminal A breast cancer surgical samples (n = 9 for each group; *p* < 0.05) (Figure 5B). In MDA-MB-231 and HCC-1937 cell lines (WT/shPANX1/shCTRL), we explored the relationship between PANX1 expression and ENTPD1/NT5E expression and found no significant correlation between PANX1 expression and ENTPD1/NT5E expression at the cell line level (*p* > 0.05; Turkey’s test; Appendix A). The above results suggest that the correlation between PANX1 expression and ENTPD1/NT5E expression at the tissue level may be due to the recruitment of immune cells with high expression of ENTPD1/NT5E in TME. The correlation between PANX1 expression and TIIC infiltration in TCGA-BRCA basal-like subtype was analyzed by TIMER. The results indicated that PANX1 expression was positively correlated with the infiltration level of neutrophils, regulatory T cells (Tregs), M2-like macrophages, and myeloid-derived suppressor cells (MDSCs) (*p* < 0.05) and was negatively correlated with the infiltration level of CD8^+^ T cells and NK cells (*p* < 0.05) (Figure 5C) in basal-like breast cancer. For HER2-enriched and Luminal B subtypes, PANX1 expression was also positively correlated with the infiltration level of neutrophils (*p* < 0.05), but it was negatively correlated with the infiltration level of CD8^+^ T cells and NK cells only in HER2 enriched subtype (*p* < 0.05) (Figure 5C). The single-cell transcriptome data revealed that TNBC tumor samples with high PANX1 expression had lower infiltration of B cells, CD4^+^ T cells, CD8^+^ T cells, myeloid cells, NK cells, and NK T cells and higher infiltration of cancer-associated fibroblasts (CAFs), plasma cells, and Tregs (Appendix A). The above results suggested that PANX1 might be a key gene responsible for exADO accumulation and establishment of an immunosuppressive TME in basal-like breast cancer.

## 4. Discussion

Malignant tumors are known as “never healing wounds”, and chronic inflammation is one of the key features of malignancy. Chronic inflammatory processes are involved in tumorigenesis and tumor progression. Purine nucleosides (ATP and ADO) exert a strong immunomodulatory ability in TME, and exATP/exADO can regulate local immune responses by activating immune cell purinergic P2 receptors. This study suggested that PANX1 was highly expressed in basal-like breast cancer and might be a poor prognostic factor. High PANX1 expression was associated with high TANs infiltration. PANX1 might play an important role in promoting TAN infiltration by increasing exATP levels. TANs could highly express ENTPD1/NT5E, which synergistically contributes to an immunosuppressive environment with high exADO levels in basal-like breast cancer.

In this study, we investigated the effect of PANX1 in basal-like breast cancer primary lesions. We found that PANX1 was highly expressed in basal-like breast cancer and could increase the exATP level in the TME and that high PANX1 expression was associated with poor prognosis in basal-like breast cancer. PANX1 knock down and probenecid (PRB) treatment reduced the levels of exATP and exADO. Although the relationship between PANX1 and exATP/exADO has been reported [38,39], the immunomodulatory role of PANX1 in the tumor microenvironment still requires further investigation. It was reported that PANX1 had certain effect on the prognosis and metastasis of various tumor, such as pro-carcinogenic effects in pancreatic adenocarcinoma [40], breast cancer [13,41], hepatocellular carcinoma [42], testicular carcinoma [43], melanoma [44], and anticarcinogenic effects in rhabdomyosarcoma [45]. Stewart et al. found that PANX1 expression was required for breast development during lactation and that high PANX1 expression was associated with worse clinical outcomes in breast cancer [13]. However, previous studies were mainly focused on the effects of PANX1 on breast cancer tumor cells [13,41], and the role of PANX1 in the formation of the immunosuppressive microenvironment has not been fully explored. Our study first proposed that the high expression of PANX1 might be one of the reasons for high infiltration of immune cells but local immunosuppression in basal breast cancer, indicating PANX1 might play an important role in constructing pro-tumor TME.

In addition, the key target cells of PANX1 in breast cancer TME need to be further explored. We found that the TANs infiltration was significantly higher in PANX1 high expression basal-like breast cancer and that the genes coexpressed with PANX1 were related to granulocyte migration and neutrophil activation. In the early stage of tumor development, as the first immune cells to enter the tumor microenvironment, neutrophils mediate subsequent immune responses and regulatory processes [46,47]. The exATP secreted by PANX1 was considered to be an important damage-associated molecular pattern (DAMP) signal [48]. Neutrophils, as the target cells of PANX1, could respond to elevated ATP concentrations [9]. exATP could exacerbate the local immune response by mediating NLRP3 inflammasome activation and IL-1β secretion via the P2Y7 receptor (P2 × 7R) on neutrophils [49]. Moreover, exATP could also delay neutrophil apoptosis via the P2Y11 receptor (P2Y11R) [50].

This study demonstrated that TANs expressing high levels of ENTPD1/NT5E could promote the hydrolysis of exATP to exADO to aggravate the immunosuppressive TME. This result was consistent with the results of studies in lipopolysaccharide (LPS)-induced inflammatory states [51,52]. Chen et al. demonstrated that neutrophils were chemotactic to exATP and hydrolyzed exATP to exADO by NT5E to promote cell migration [52,53]. In addition, exADO inhibits neutrophil adhesion and the release of TNF-α and chemokines from LPS-stimulated neutrophils [54]. Neutrophil-expressed adenosine A2A receptor (A2AR) could inhibit the neutrophil recruitment cascade [55]. Previous studies also revealed that the purinergic receptor P2Y6 receptor (P2Y6R) [56], adenosine A2B receptor (A2BR) [57], and adenosine A3 receptor (A3R) [58] on neutrophils were involved in the regulation of neutrophil extracellular traps (NETs) in inflammatory states. Whether exATP/exADO can regulate NETs or the N1/N2-like subtype transition of TANs through purinergic receptors and further affect the development and metastasis of breast cancer requires further investigation. In addition, our study results also suggested some CCLs, such as CCL3 and CCL4 as well as vascular endothelial growth factor (VEGF) and baculoviral IAP repeat containing 5 (BIRC5), were highly expressed in TANs transcriptome analysis. CCLs were reported to have played an important role of breast tumor cell–neutrophil interactions in regulating pro-tumor characteristics in neutrophils [59,60]. VEGF was known as a primary stimulant of angiogenesis, and it was a macrophage chemotactic cytokine [61]. It was reported that VEGF level was correlated with MPO [62]. Moreover, BIRC5 was reported to play an important role in carcinogenesis by influencing cell division and proliferation and inhibiting apoptosis [63].

Our study demonstrated that ENTPD1 and NT5E expressions were higher in the PANX1 high expression basal-like breast cancer and PANX1 upregulated exADO levels in the TME. At the tissue level, there was a positive correlation between the expression of PANX1 and the expression of ENTPD1 and NT5E. At the cell line level, the expression of ENTPD1 and NT5E was not affected by the expression of PANX1. This suggests that a high expression of PANX1 may lead to high expression of ENTPD1 and NT5E at the tissue level by recruiting high expression of ENTPD1 and NT5E immune cells. PANX1 was an important immunomodulator in the TME. When combining the results of single-cell transcriptome data analysis with TCGA-BRCA data analysis in basal-like breast cancer, we found that PANX1 expression was negatively correlated with the infiltration levels of CD8^+^ T cells and NK cells, and it was positively correlated with the infiltration levels of Tregs. Higher PANX1 expression caused an increase in exATP, which was further catabolized by ENTPD1 and then converted to adenosine by NT5E [64]. Thus, the prognostic value of PANX1 was not independent of NT5E/ENTPD1. Previous studies mainly focused on the role of the nucleotidases ENTPD1 and NT5E in exADO production [65,66]. As the upstream source of exATP, PANX1 could be a potential and effective therapeutic target. Previous studies also showed that PANX1 promoted the activation of NLRP3 (NOD-, LRR- and pyrin domain-containing protein 3) inflammasome and increased the level of interleukin 1β (IL-1β) in the local microenvironment [67]. In-depth investigations of PANX1 may help elucidate tumor-inflammation interactions.

There are also some limitations in our study. First, we selected representative PAM50 basal-like subtype for analysis rather than TNBC. As a group with strong heterogeneity, TNBC can be further divided into more subtypes. As the subtype with the highest proportion in TNBC, studying the TME characteristics of basal-like subtype could help deepen our understanding of TNBC. Second, although our study results indicated that high PANX1 expression was closely related to high ENTPD1/NT5E expression in the basal-like breast cancer TME, we could not draw definitive conclusions on cause-effect correlations, as the sample size for verifying the bioinformatics analysis was relatively small and we did not perform in vivo experiments. However, a correlation could spawn hypotheses, which then can be tested in future studies. The cause–effect correlations need to be confirmed by further experimental validation in vivo and in vitro in the future. Third, we mainly focused on the effect of PANX1 expression levels on breast cancer TME in this study. However, as a channel protein, the structure and activity of PANX1 are crucial for its function. The effects of different structures and different activation levels of PANX1 on the state of the breast cancer TME deserve further investigation.

## 5. Conclusions

In summary, the expression of PANX1 was positively correlated with TANs infiltration through exATP secretion in basal-like breast cancer. The high expression of ENTPD1/NT5E in TANs could synergistically establish an immunosuppressive TME with high exADO levels. In this study, the relationship between high exATP/exADO levels and TANs was investigated to elucidate the properties of PANX1 and its ability to reshape the metabolic-immunosuppressive TME and provide new targets and strategies for breast cancer treatment.

## Figures and Tables

**Figure 1 cancers-14-03369-f001:**
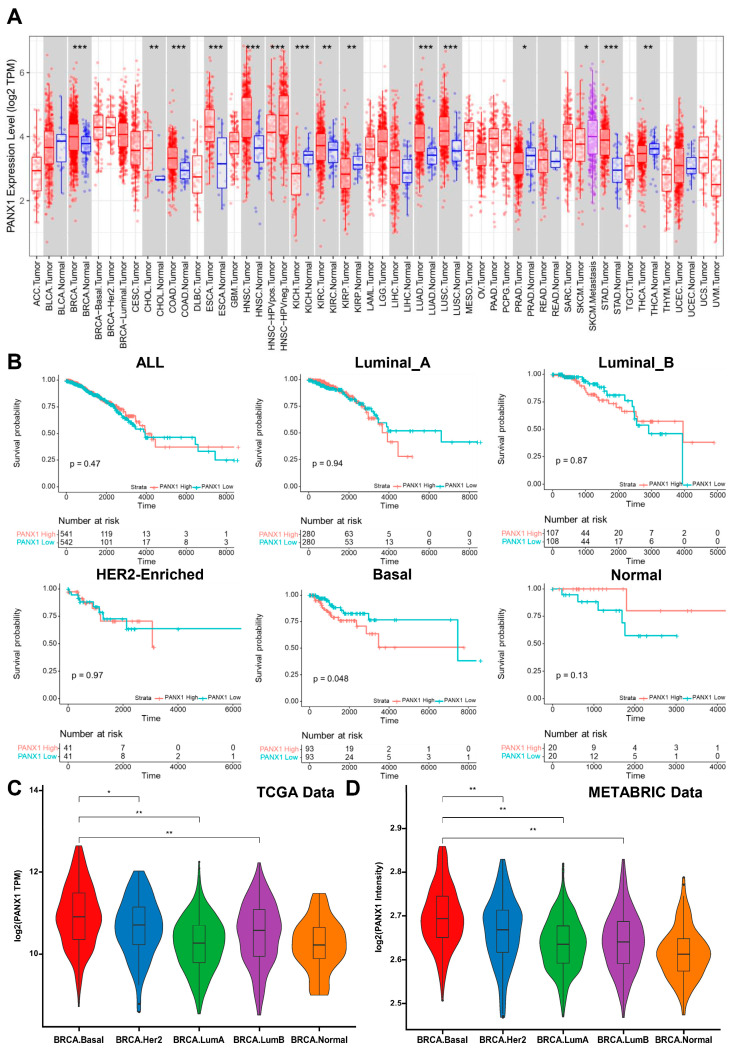
(**A**) PANX1 was highly expressed in breast cancer (BRCA), colon adenocarcinoma (COAD), esophageal carcinoma (ESCA), head and neck squamous cell carcinoma (HNSC), kidney chromophobe (KICH), lung adenocarcinoma (LUAD), lung squamous cell carcinoma (LUSC), and stomach adenocarcinoma (STAD) compared with normal tissues (*p* < 0.001 as significant; Student’s *t* test) (TCGA-BRCA data; red bar: tumor tissue; blue bar: normal tissue); (**B**) correlation between PANX1 expression and overall survival (OS) in breast cancer under PAM50 molecular intrinsic subtypes (All (n = 1083), Luminal A (n = 560), Luminal B (n = 215), HER2-enriched (n = 82), Basal (n = 186), and Normal (n = 40)); (**C**,**D**) PANX1 expression under different PAM50 breast cancer molecular intrinsic subtype (TCGA-BRCA (n = 1083) and METABRIC (n = 1699) data, PAM50 algorithm; Turkey’s test) (* *p* < 0.05; ** *p* < 0.01; *** *p* < 0.001).

**Figure 2 cancers-14-03369-f002:**
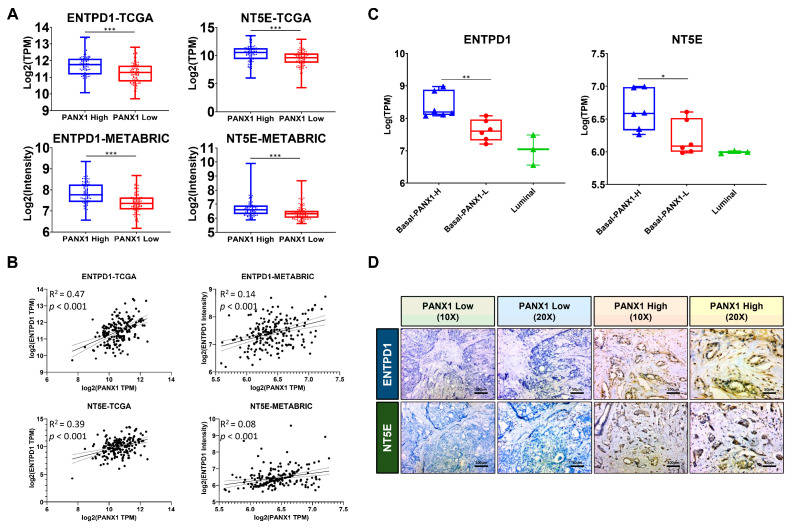
(**A**) ENTPD1 and NT5E expression in basal-like breast cancer under high and low level of PANX1 expression (*p* < 0.01) (TCGA-BRCA (n = 186) and METABRIC (n = 199) basal-like subtype data; Student’s *t* test); (**B**) ENTPD1 and NT5E expression was positively correlated with PANX1 expression in basal-like breast cancer (*p* < 0.001) (TCGA-BRCA and METABRIC basal-like subtype data; Pearson’s correlation); (**C**) ENTPD1 and NT5E expression in breast cancer specimens (Basal-like PANX1 high subgroup: n = 6; basal-like PANX1 low subgroup: n = 6; Luminal subtype: n = 3) (*p* < 0.05; Turkey’s test); (**D**) ENTPD1 and NT5E expression in basal-like breast cancer surgical specimens with different PANX1 expression levels by immunohistochemistry at 10× and 20× magnifications (* *p* < 0.05; ** *p* < 0.01; *** *p* < 0.001).

**Figure 3 cancers-14-03369-f003:**
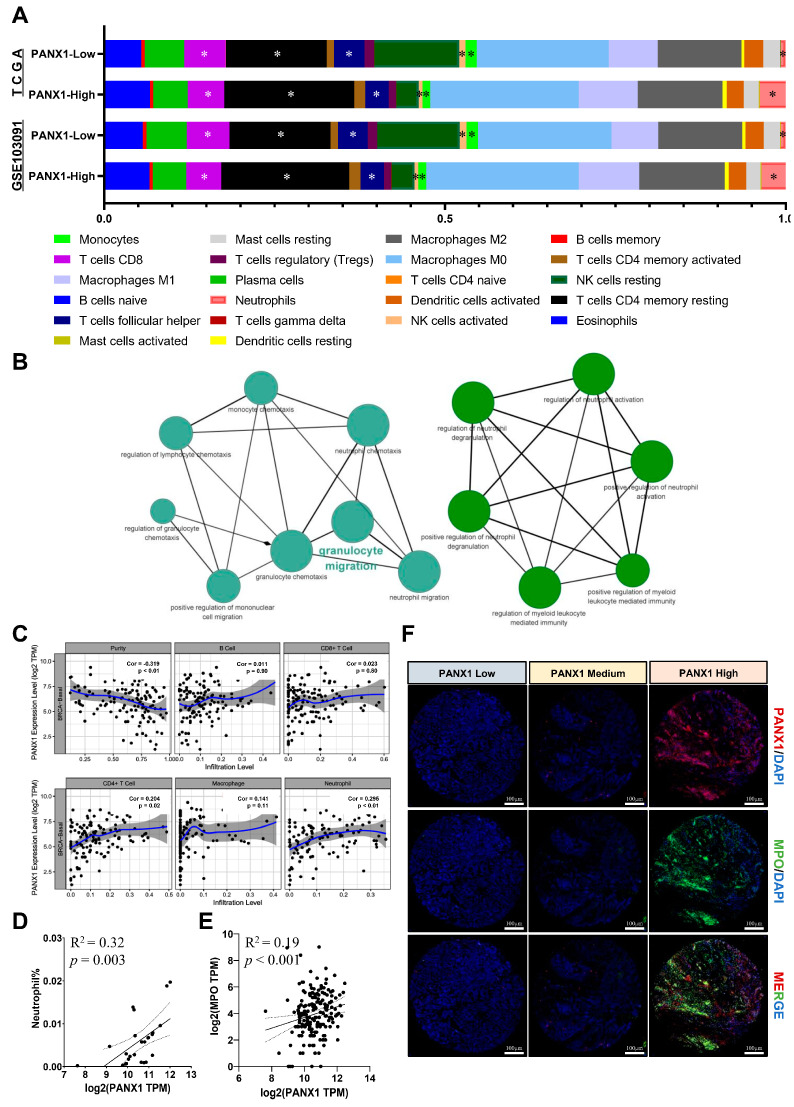
(**A**) CIBERSORT evaluated tumor-infiltrating immune cell (TIIC) differences under high (top 50%)/low (bottom 50%) PANX1 expression in basal-like subtype breast cancer (The key difference immune cells are shown by star (* *p* < 0.05); TCGA-BRCA basal-like subtype (n = 186) and GSE103091(n = 238) data); (**B**) GO analysis of PANX1 and its coexpression immune-related genes; (**C**) TIMER analysis of PANX1 expression and TIIC correlation (TCGA-BRCA data; n = 1083; Spearman’s rank correlation); (**D**) neutrophil abundance was positively correlated with PANX1 expression in METABRIC basal-like subtype data (R^2^ = 0.32; *p* = 0.003; Pearson’s correlation) (CIBERSORT algorithms; n = 25, outliers have been filtered); (**E**) MPO (neutrophil marker) expression was positively correlated with PANX1 expression in TCGA-BRCA basal-like subtype data (n = 186; R^2^ = 0.19; *p* < 0.001; Pearson’s correlation); (**F**) immunofluorescence detection of PANX1/MPO coexpression in basal-like breast cancer paraffin-embedded pathological specimens (DAPI, 4′,6-diamidino-phenylindole; MPO, myeloperoxidase).

**Figure 4 cancers-14-03369-f004:**
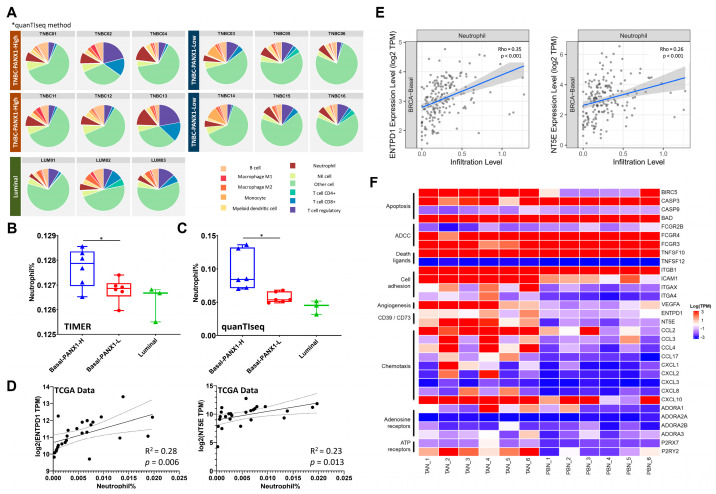
(**A**) The proportion of infiltrating TANs in Luminal (n = 3) and basal-like subtype (High PANX1 expression: n = 6; low PANX1 expression: n = 6) surgical specimens; (**B**,**C**) basal-like breast cancer with high PANX1 expression had more infiltrating TANs than basal-like breast cancer with low PANX1 expression and the Luminal subtype (*p* < 0.05 for TIMER; *p* = 0.11 for quanTIseq; Student’s *t* test); (**D**) the correlation between ENTPD1/NT5E expression and TAN infiltration in basal-like breast cancer (n = 25; *p* < 0.05; R^2^ = 0.28 (ENTPD1) and 0.23 (NT5E); Pearson’s correlation; TCGA-BRCA data; CIBERSORT-LM22 algorithms; outliers have been filtered); (**E**) TIMER analysis suggested a positive correlation between ENTPD1/NT5E expression and TAN infiltration in the basal-like subtype (n = 186; *p* < 0.01; Rho = 0.35 (ENTPD1) and 0.26 (NT5E); Spearman’s rank correlation; TCGA-BRCA data); (**F**) heatmap of the transcriptome analysis of TANs (n = 6) and PBNs (n = 6) in basal-like breast cancer (TANs, tumor-associated neutrophils; PBNs, peripheral blood neutrophils) (* *p* < 0.05).

**Figure 5 cancers-14-03369-f005:**
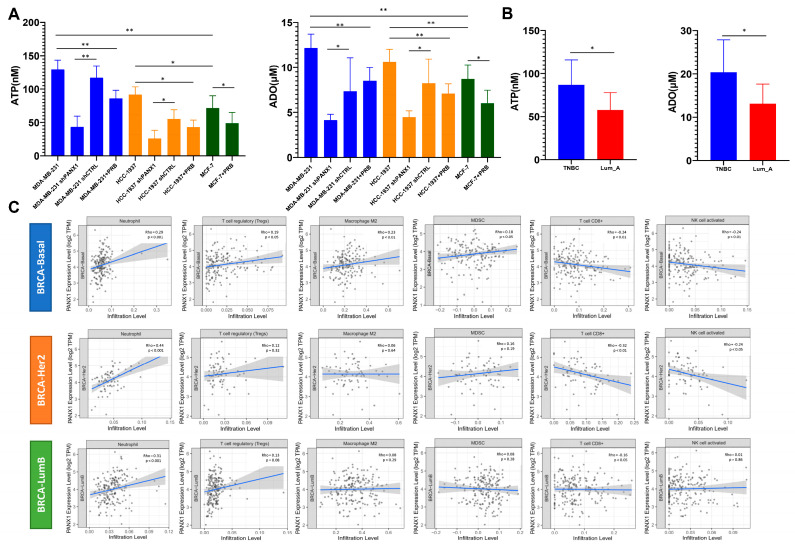
(**A**) Levels of exATP and exADO in MDA-MB-231, HCC-1937 and MCF-7 cell culture media; PANX1 knock down and probenecid (PRB) treatment reduced the levels of exATP and exADO in the supernatant of MDA-MB-231 and HCC-1937 cells (n = 19 for each group; *p* < 0.05; Student’s *t* test); (**B**) levels of exATP and exADO in the supernatant of digested tissue from triple-negative and Luminal A breast cancer surgical specimens (*p* < 0.05; n = 9 for each group; Student’s *t* test); (**C**) the correlation between PANX1 expression and infiltration levels of neutrophils, Tregs, M2-like macrophages, MDSC cells, CD8^+^ T cells, and NK cells in the tumor microenvironment for Luminal B, HER2 enriched and basal-like breast cancer by TIMER (TCGA-BRCA data; Spearman’s rank correlation). (* *p* < 0.05; ** *p* < 0.01).

**Table 1 cancers-14-03369-t001:** Clinical baseline characteristics of the included patients and corresponding experimental procedures for the specimens.

No.	Age	ER	PR	HER2	Ki-67	Subtype	WHO Grade	Stage	Stromal TILs%	Bulk RNA-Seq	PAM50	Barcode	IHC	IF	TAN RNA-seq	Paired PBN RNA-seq	ATP/ADO Assay
PT01	50	Neg	Neg	Neg	20%	TNBC	3	IIA	5.0%		NA						Yes
PT02	37	Neg	Neg	Neg	70%	TNBC	3	IIA	30.0%		NA						Yes
PT03	41	Neg	Neg	Neg	40%	TNBC	3	IIIC	7.0%		NA						Yes
PT04	48	Neg	Neg	1+	50%	TNBC	3	IIB	30.0%	Yes	Basal-like	TNBC08					Yes
PT05	62	Neg	Neg	Neg	40%	TNBC	3	IIA	65.0%	Yes	Basal-like	TNBC02	Yes	Yes	Yes	Yes	
PT06	61	Neg	Neg	1+	30%	TNBC	3	IIA	10.0%	Yes	Basal-like	TNBC07		Yes	Yes	Yes	
PT07	44	Neg	Neg	Neg	70%	TNBC	3	IIA	20.0%		NA						Yes
PT08	47	Neg	Neg	Neg	70%	TNBC	3	IIA	7.0%		NA						Yes
PT09	54	Neg	Neg	1+	15%	TNBC	2	IIA	3.0%	Yes	Basal-like	TNBC06	Yes				
PT10	48	Neg	Neg	Neg	15%	TNBC	3	I	60.0%	Yes	Basal-like	TNBC05			Yes	Yes	
PT11	59	Neg	Neg	Neg	30%	TNBC	2	I	10.0%	Yes	Basal-like	TNBC09			Yes	Yes	
PT12	53	Neg	Neg	1+	40%	TNBC	3	IIA	40.0%	Yes	Basal-like	TNBC03	Yes				
PT13	61	Pos	Pos	1+	8%	Luminal A	2	IIA	5.0%		NA						Yes
PT14	59	Pos	Pos	1+	5%	Luminal A	2	IIA	5.0%		NA						Yes
PT15	87	Pos	Pos	1+	5%	Luminal A	2	IIA	5.0%		NA						Yes
PT16	61	Pos	Pos	1+	10%	Luminal A	3	IIA	5.0%	Yes	Luminal-A	LUM03					
PT17	47	Pos	Pos	1+	10%	Luminal A	3	IIB	4.0%		NA						Yes
PT18	51	Pos	Pos	1+	15%	Luminal A	2	IIA	10.0%	Yes	Luminal-A	LUM01					
PT19	66	Pos	Pos	1+	5%	Luminal A	2	I	5.0%		NA						Yes
PT20	61	Pos	Pos	Neg	5%	Luminal A	2	IIA	7.0%		NA						Yes
PT21	73	Pos	Pos	1+	15%	Luminal A	3	IIA	5.0%	Yes	Luminal-A	LUM02					
PT22	61	Pos	Pos	Neg	5%	Luminal A	2	IIA	2.0%		NA						Yes
PT23	76	Pos	Pos	Neg	10%	Luminal A	3	IIA	2.0%		NA						Yes
PT24	52	Pos	Pos	Neg	10%	Luminal A	2	IIA	5.0%		NA						Yes
PT25	55	Neg	Neg	Neg	40%	TNBC	2	IIA	55.0%	Yes	Basal-like	TNBC04	Yes		Yes	Yes	Yes
PT26	35	Neg	Neg	1+	10%	TNBC	2	IIB	20.0%	Yes	Basal-like	TNBC10	Yes	Yes			Yes
PT27	52	Neg	Neg	Neg	15%	TNBC	2	IIA	12.0%	Yes	Basal-like	TNBC01	Yes		Yes	Yes	Yes
PT28	45	Neg	Neg	Neg	70%	TNBC	3	IIB	25.0%	Yes	Basal-like	TNBC11					
PT29	51	Neg	Neg	Neg	65%	TNBC	3	I	45.0%	Yes	Basal-like	TNBC12					
PT30	28	Neg	Neg	Neg	60%	TNBC	3	IIA	50.0%	Yes	Basal-like	TNBC13					
PT31	60	Neg	Neg	Neg	75%	TNBC	3	IIB	30.0%	Yes	Basal-like	TNBC14					
PT32	71	Neg	Neg	Neg	45%	TNBC	3	IIB	10.0%	Yes	Basal-like	TNBC15					
PT33	43	Neg	Neg	Neg	50%	TNBC	2	IIA	7.0%	Yes	Basal-like	TNBC16					

ER: estrogen receptor; PR: progesterone receptor; HER2: human epidermal growth factor receptor 2; IHC: immunohistochemistry; IF: immunofluorescence; TAN: tumor-associated neutrophil; PBN: peripheral blood neutrophil.

**Table 2 cancers-14-03369-t002:** Clinical characteristics of the included samples for immunohistochemistry.

No.	Age	Subtypes	Ki-67	WHO Grade	Stage	Stromal TILs%	Bulk RNA-Seq	PAM50	Barcode	IHC	IHC-PANX1
PT05	62	TNBC	40%	3	IIA	65.0%	Yes	Basal-like	TNBC02	Yes	High
PT25	55	TNBC	40%	2	IIA	55.0%	Yes	Basal-like	TNBC04	Yes	High
PT27	52	TNBC	15%	2	IIA	12.0%	Yes	Basal-like	TNBC01	Yes	High
PT09	54	TNBC	15%	2	IIA	3.0%	Yes	Basal-like	TNBC06	Yes	Low
PT12	53	TNBC	40%	3	IIA	40.0%	Yes	Basal-like	TNBC03	Yes	Low
PT26	35	TNBC	10%	2	IIB	20.0%	Yes	Basal-like	TNBC10	Yes	Low

TILs: tumor infiltrating lymphocytes; IHC: immunohistochemistry.

## Data Availability

The data from previously reported studies and datasets, which support this research, have been cited in the manuscript. All raw data and results generated during and/or analyzed during the current study have been publicly deposited in the Zenodo repository and can be accessed at https://doi.org/10.5281/zenodo. 6806339.

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
