# Peer review of "High PANX1 Expression Leads to Neutrophil Recruitment and the Formation of a High Adenosine Immunosuppressive Tumor Microenvironment in Basal-like Breast Cancer"

_cancers, 2022, doi:10.3390/cancers14143369_

Round 1

Author Response

Dear Editors and Reviewers

Thank you very much for your kindly comments and suggestions on our manuscript. We carefully revised the manuscript based on your suggestions. We are now sending the revised article for your re-consideration. Point to point responses have been made for all your suggestions in the attachment. Thank you again for your time and consideration.

Reviewer 2 Report

Summary

The manuscript by Chen and colleagues examines PANX1 expression in breast cancer. Initially the authors establish that PANX1 expression is increased in basal-type breast cancer specimen, which also show decreased survival. There was correlation between PANX1 expression and ENTPD1 and NT5E expression in basal-type breast cancer specimen. Consequently, infiltrating immune cell patterns were analyzed between high-PANX1 and low-PANX1 expressing basal-type breast tumors and it was found that infiltrating neutrophil and CD4 T-cell amount was higher in high-PANX1 expressing basal-type breast cancer. In surgical specimen TAN infiltration was higher in high-PANX1 expressing basal-type breast cancers than in other subtypes. TAN infiltration was positively correlated with ENTPD1 and NT5E expression in basal-type breast cancers. Lastly the authors demonstrate that extracellular ATP and ADO levels are lower in PANX1-knockdown breast cancer cells.

General comments

There is an interesting finding that PANX1 expression correlated with higher ENTPD1/NT5E expression. The authors mention that they lacked the data to verify cause-effect relations. Since this relationship was only observed in patient samples, it leaves the door open that the two effects are not related and induced by an external tertiary factor. Would the authors consider measuring expression of ENTPD1/NT5E in their PANX1 knockdown cells in order to provide some evidence that PANX1 expression and ENTPD1/NT5E expression is causally related?

Figure 3A demonstrates the correlation of TANs with PANX1 expression. However, it also shows that high PANX1 expressing tumors show higher CD4+ and lower CD8+ expression. Since there is some evidence that immune cell infiltration is correlated with tumor stage, could there be a possibility that PANX1 expression is correlated with tumor stage?

Some figures have small writing and/or insufficient pixelation.

Specific comments

It is somewhat unclear how the authors identified PANX1 as a potentially dysregulated protein in cancer cells from the introduction. Perhaps the Steward study (42) could be placed in the introduction to provide context for the investigation.

For 2.4 RNA sequencing, could the authors clarify that they extracted RNA as the text indicates they extracted DNA.

There needs to be some data demonstrating PANX1 knockdown following shRNA treatment. This could be a supplemental figure.

Author Response

(The authors gave the same response as above.)
